# Agile Flight with Optimization Embedded Networks

## Abstract

To bridge the gap between perception and planning in traditional navigation systems, we address the challenge of learning optimal trajectories directly from depth information in an end-to-end fashion. Using neural networks as black-box replacements for traditional modules can compromise robustness and stability. Moreover, such methods often fail to adequately account for the robot's kinematic constraints, leading to trajectories that may not be satisfactorily executable. In this paper, we integrate the strengths of conventional methods and neural networks by introducing an optimization-embedded neural network based on a compact trajectory library. Neural networks establish spatial constraints for model-based trajectory planning, followed by robust numerical optimization to achieve feasible and optimal solutions. By making the process differentiable, our model seamlessly approximates the optimal trajectory. Additionally, the introduction of a regularized trajectory library enables the method to efficiently capture the spatial distribution of optimal trajectories with minimal storage cost, ensuring multimodal planning characteristics. Evaluations in complex, unseen environments demonstrate our method's superior performance over state-of-the-art algorithms. Real-world flight experiments with a small onboard computer showcase the autonomous quadrotor's ability to navigate swiftly through dense forests.

## 1 Introduction

Unmanned aerial vehicles (UAVs) (Fan et al., 2020; Muchiri & Kimathi, 2022), owing to their compact and straightforward hardware structure combined with agile and high maneuverability, have found extensive applications across various fields such as aerial photography, exploration, and search and rescue operations. As a critical component for achieving these tasks, efficient and robust autonomous navigation modules have garnered significant attention from both industry and academia. Traditionally, navigation modules employ sensors like depth cameras to perceive the environment,explicitly constructing occupancy maps and computing environment representations favorable for motion planning, such as Euclidean signed distance fields (ESDF) (Oleynikova et al., 2016; Han et al., 2019; Reijgwart et al., 2019; Finean et al., 2021) or neural radiance fields (NeRF) (Adamkiewicz et al., 2022; Pantic et al., 2022; Lin & Yi, 2022). Subsequently, motion planning algorithms based on search or optimization are utilized on these maps to compute optimal trajectories that account for initial and final states as well as obstacle avoidance. Although intuitive from an engineering perspective, this modular decomposition framework inevitably introduces additional physical delays. Additionally, this modular approach often leads to a lack of cohesion between sub-modules and requires extensive manual parameter tuning by engineers. Recently, learning-based navigation (Tang et al., 2018; Xiao et al., 2022; Dong et al., 2023) has gained widespread attention due to its efficient integration of perception and planning modules. This end-to-end approach directly outputs trajectories from raw sensor data, bypassing explicit mapping. However, this methodology heavily relies on the neural network's capabilities, leading to a black-box system that poses challenges for debugging and reduces the system's extensibility and interpretability. Furthermore, given the constraints of physical platforms or specific task scenarios, there is a need to impose various custom constraints on the trajectory, such as the kinematic constraints of the robot. This imposes significant challenges on the network. Researchers often devise more complex strategies for the network to address these constraints, but these strategies can sometimes compromise optimality or fail to ensure constraint satisfaction adequately, affecting the overall system's completeness.

To address the limitations of existing methods, this work integrates traditional trajectory optimization with neural networks to create an end-to-end visual navigation system that directly generates kinematically feasible spatial-temporal optimal trajectories from depth images without explicit mapping. Compared to conventional learning-based motion planning algorithms, our approach is distinguished by the use of implicit differentiation to embed the trajectory optimization within the neural network, enabling coupled training. This method ensures optimality while alleviating the burden on the network, enhancing the system's interpretability and extensibility, and fundamentally guaranteeing the feasibility of kinematic constraints. Specifically, rather than naively producing a series of trajectory points, our neural network extracts safe guiding regions from depth images and reconstructs them as geometric space constraints for trajectory optimization, which then incorporates user-specified kinematic constraints to efficiently and robustly converge ($\sim 1ms$) to high-quality trajectories. Making numerical optimization differentiable allows it to be modeled as a layer within the neural network, enabling direct backpropagation of the trajectory evaluation loss gradient and encouraging the network to focus on regions yielding optimal trajectories. To fully explore the environment and align the multimodal nature of local motion planning, our neural network outputs a mixture distribution over an offline regularized lightweight motion primitive library. Based on the probabilities, specific motion primitives are selected and allocated safe feasible spaces, which are subsequently input into the optimization module to generate maneuverable and agile motion. We conduct extensive comparative experiments against various state-of-the-art motion planning algorithms across multiple scenarios, including both traditional and learning-based methods. The results demonstrate that our method has significant advantages in success rate, optimality, and constraint satisfaction. Moreover, as shown in Figure 1. B, we carry out real-world experiments to validate the practical applicability of our algorithm on fully autonomous physical platform without relying on external perception and localization.

## 2 RELATED WORK

### 2.1 TRADITIONAL MOTION PLANNING

Search or sampling-based algorithms in discrete configuration spaces (Pivtoraiko & Kelly, 2005; Webb & Van Den Berg, 2013; Klemm et al., 2015) are prevalent in robotic motion planning, integrating custom constraints and avoiding local optima. However, they require numerous samples for high-quality trajectories, impairing real-time efficiency. Conversely, optimization-based methods (Zhou et al., 2019; Gao et al., 2020; Zhou et al., 2020; Tordesillas & How, 2021) leverage gradient information to efficiently converge to feasible trajectories in continuous spaces, balancing quality and time, and becoming mainstream for UAV local planners. These methods often require explicit environmental modeling through depth information and manual safety constraint extraction, for example, using ESDF to construct safety constraints (Zhou et al., 2021; 2019). However, constructing ESDF incurs additional computational costs and involves a trade-off between efficiency and accuracy. The ego planner (Zhou et al., 2020) avoids ESDF construction by iteratively generating safe guidance paths for obstacle avoidance gradients, but it lacks convergence guarantees and may get trapped in unsafe local minima in complex environments. Using guidance paths to deform trajectories can also deviate from the original optimization problem, affecting optimality. Corridor-based methods (Gao et al., 2020; Tordesillas & How, 2021) gain popularity in local motion planning by extracting feasible convex hulls from environment point clouds using geometric computations to model safety constraints. However, they require a collision-free path to seed the convex hull, often obtained using low-dimensional algorithms like A*(Choset, 2007) or hybrid A*(Ding et al., 2018), which typically do not consider the robot's higher-order kinematics, resulting in convex hulls that are not conducive to generating agile trajectories.

### 2.2 LEARNING-BASED MOTION PLANNING

Learning-based methods (Loquercio et al., 2021; Allen & Pavone, 2019; Chou et al., 2021; Yang et al., 2023; Roth et al., 2023; Kulkarni & Alexis, 2024; Jacquet & Alexis, 2024; Han et al., 2024; Wu et al., 2024) emerge as promising approaches in local planning, eliminating the need for explicit mapping and reducing latency. Loquercio et al. (2021) leverage deep convolutional neural networks to learn flight trajectories from depth images, using human pilot trajectories as supervision. However, this method requires high-quality and large-scale datasets. Recently, some approaches combine

networks with numerical optimization. For instance, Jacquet & Alexis (2024) learn collision probabilities for any point in space using a network, which are then modeled as safety constraints in trajectory optimization. Similarly, Han et al. (2024) address finer obstacle avoidance by modeling the robot's shape as a convex hull and predicting the signed distance between the hull and the nearest obstacle using a neural network. Although these works integrate networks and optimization, they differ from our approach in that the network and optimization are independent components. In contrast, our algorithm incorporates differentiable optimization as part of the network training process, facilitating the evolution of the network in directions beneficial for subsequent optimization and generating higher-quality solutions while ensuring maneuverability and agile flight. Some works also apply the concept of bilayer optimization. Chen et al. (2024) propose the IA* algorithm, which extends the traditional A* search algorithm by embedding it within a neural network for training. However, the paths generated by this algorithm consist of discrete grid points, rendering it unsuitable for direct tracking by high-speed drones. Wu et al. (2024) utilize LSTM(Graves & Graves, 2012) to learn time allocation for piecewise polynomial trajectories, achieving optimal solutions under ideal conditions. This method, however, requires a known global map and offline convex decomposition of safe regions, making it impractical for vision-based end-to-end local planners. Iplanner (Yang et al., 2023) is an end-to-end local planner that generates trajectories from depth images, validated by numerous real-world experiments. This approach uses closed-form cubic splines to interpolate the points output by the network, approximating a feasible solution to the original trajectory planning problem. Despite being lightweight, this approximation does not strictly guarantee adherence to dynamic constraints, potentially resulting in trajectories that are difficult for the physical platform to execute during high-speed flight, leading to crashes.

## 3 SPATIAL-TEMPORAL TRAJECTORY OPTIMIZATION FORMULATION

The objective of visual navigation is to find a dynamically feasible trajectory within a safe region $\mathcal{E}_{free} \in \mathbb{R}^{H \times W}$, guided by depth observations $\boldsymbol{D} \in \mathbb{R}^{H \times W}$, while satisfying initial and terminal state constraints. In this paper, the trajectory $\xi(t) : [0, T] \to \mathbb{R}^3$ is represented using MINCO (Wang et al., 2022), a special piecewise polynomial that adheres to the principle of minimum energy, and is parameterized by trajectory duration $T \in \mathbb{R}^+$ and a series of waypoints $\boldsymbol{Q} = [\boldsymbol{q}_1, ..., \boldsymbol{q}_{N-1}] \in \mathbb{R}^{3 \times (N-1)}$. $N$ is the number of pieces of the trajectory. With this compact representation, our trajectories inherently satisfy the initial and terminal state conditions and ensure high-order continuity at waypoints between adjacent polynomials. These features formally lay a solid foundation for smooth and coherent motion. Subsequently, as shown in Figure 1. A, we use flight corridors $\mathcal{F}$ to represent free space, which are modeled as safety constraints to restrict the shape of the trajectory space. Consequently, the trajectory optimization that minimizes control energy and includes first-order time regularization is formulated as follows:

$$\min_{\xi(\boldsymbol{Q},T),\mathcal{F}} J = \int_0^T (\upsilon(t))^{\mathrm{T}} \boldsymbol{W} \upsilon(t) dt + \rho T \tag{1}$$

$$s.t. \quad \mathcal{C}(\xi(t), \xi^{(1)}(t), ..., \xi^{(u)}(t)) \leq 0, \forall t \in [0, T], \tag{2}$$

$$||\xi(t) - \zeta(t)||_2^2 \leq \gamma^2(t), \forall t \in [0, T], \tag{3}$$

$$\mathcal{F}(\zeta, \gamma)(t) \in \mathcal{E}_{free}, \forall t \in [0, T], \tag{4}$$

$$\upsilon(t) = \xi^{(u)}(t), \forall t \in [0, T], \tag{5}$$

where $\boldsymbol{W} \in \mathbb{R}^{3 \times 3}$ is a positive diagonal matrix to penalize control efforts and $\rho \in \mathbb{R}^+$ is the time regularization. $u$ denotes the derivative order of the control input $\upsilon$ with respect to the trajectory $\xi$. $\mathcal{C}$ generally refers to various user-defined constraints tailored to specific task scenarios, such as constraints on velocity, acceleration, thrust, torque, and others. In this paper, $\mathcal{F}$ represents the flight corridors, modeled as spheres with centers at $\zeta$ and radii $\gamma$. Eq. (3) serves as a spatial constraint to confine the trajectory within the safe corridors. In traditional algorithms (Gao et al., 2020; Tordesillas & How, 2021) , it is worth noting that flight corridors are typically predetermined and do not participate in the optimization process. These algorithms usually involve using a frontend path planning algorithm to find a path on a grid map, followed by computational geometry methods to derive the corridors along this path. However, this extensive modular layering introduces latency and reduces the optimality of the solution. To ensure efficiency, the cost function of the frontend path planning algorithm often differs from the final trajectory's evaluation metrics, lead-

Figure 1: Quadrotor flight in forests. The top figure illustrates the visualization of the trajectory and corridors. The bottom figure showcases the real high-speed autonomous flight of the quadrotor in the wild.

ing to corridors that may not be conducive to subsequent trajectory optimization. Additionally, this explicit decoupling strategy eliminates the possibility of dynamically adjusting the corridors based on the optimized trajectory quality. In this paper, we utilize neural networks to learn corridors in an end-to-end manner. By integrating coupled training with trajectory optimization, as discussed in subsequent sections, we directly propagate the gradient related to trajectory quality back to the corridor network, ensuring adaptive dynamic adjustment.

## 4 NETWORK ARCHITECTURE

### 4.1 END-TO-END NAVIGATION SYSTEM OVERVIEW

The planning pipeline is illustrated in Figure 2. Our method addresses a bilevel optimization form that is equivalent in optimality to the original optimization problem Eq. (1-5):

$$\min_{\phi} J = J(\xi^*(\boldsymbol{Q}^*, T^*)) \tag{6}$$

$$s.t. \quad \mathcal{F}_{\phi}(\zeta, \gamma) \in \mathcal{E}_{free}, \tag{7}$$

$$\xi^*(\boldsymbol{Q}^*, T^*) = \arg\min_{\xi(\boldsymbol{Q}, T)} J = J(\xi(\boldsymbol{Q}, T)), s.t.Eq.(2-5)(\phi), \tag{8}$$

where $\phi$ is the parameters of the neural network and $\xi^*$ denotes the optimal trajectory derived from the objective $J$ within the given spatial constraints. Intuitively, the model-free neural network undertakes the outer optimization, extracting the corridor from the depth information, while the model-based trajectory planning performs the inner optimization, determining the spatial-temporal optimal trajectory. After obtaining the trajectory, the corresponding metric will backpropagate through the differentiable trajectory optimization layer, thereby updating the network's parameters. This synergistic process of network updating and trajectory optimization constitutes a nested, coupled bilevel optimization framework, enabling the network to directly output the safe regions most conducive to trajectory generation. Moreover, this bilevel optimization Eq. (6-8) formulation does not sacrifice optimality in principle, as demonstrated in the Appendix A, it shares the same optimal solution as the original optimization problem Eq. (1-5).

## 4.2 SAFE SPACE EXTRACTION LAYER

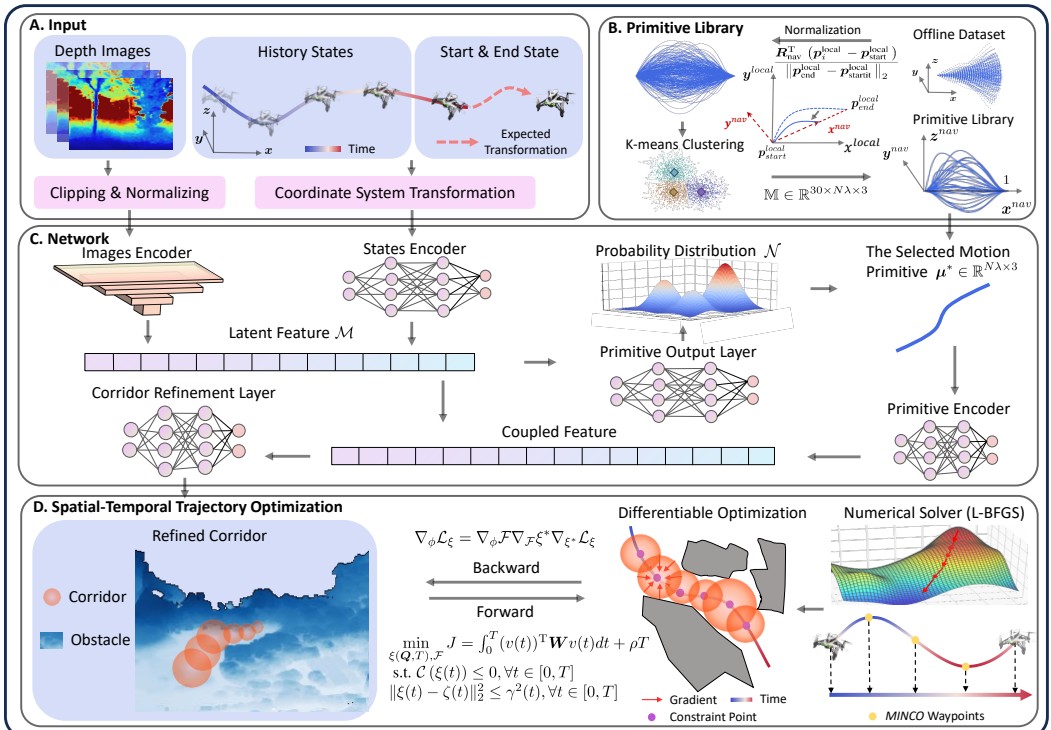

Figure 2: Planning pipeline.

Before delving into the specific details of the network structure, we first instantiate its output representation. To address the time-continuous constraint Eq. (2, 3), similar to the work (Han et al., 2023), we discretize each piece $\xi_i(t) : [0, \frac{T}{N}] \to \mathbb{R}^3$ of the piecewise polynomial trajectory into $\lambda$ constraint points. By imposing constraints at these discrete points, we can effectively control the entire trajectory:

$$\mathcal{C}(\xi(t), \xi^{(1)}(t), ..., \xi^{(u)}(t)) \leq 0, \forall t \in [0, T],$$

$$||\xi(t) - \zeta_\phi(t)||_2^2 \leq \gamma_\phi^2(t), \forall t \in [0, T] \iff$$

$$\mathcal{C}(\xi_i(\frac{jT}{\lambda N}), \xi_i^{(1)}(\frac{jT}{\lambda N}), ..., \xi_i^{(u)}(\frac{jT}{\lambda N})) \leq 0, \forall i \in \{1, ..., N\}, \forall j \in \{1, ..., \lambda\}, \tag{9}$$

$$||\xi_i(\frac{jT}{\lambda N}) - \zeta_{\phi,i,j}||_2^2 \leq \gamma_{\phi,i,j}^2, \forall i \in \{1, ..., N\}, \forall j \in \{1, ..., \lambda\}. \tag{10}$$

The physical significance of Eq. (10) lies in the fact that the neural network needs to assign a safety sphere parameterized by $\zeta_{\phi,i,j}, \gamma_{\phi,i,j}$ to each constraint point $\xi_{i,j} = \xi_i(\frac{jT}{\lambda N})$ along the trajectory. Consequently, the network is required to output a total of $N\lambda$ spheres. Moreover, we would like to emphasize that the network has the theoretical capability to output convex hulls of arbitrary shapes. However, for the sake of simplicity and ease of understanding, we use spheres as the representation form for the corridor.

To achieve the aforementioned goals, we design a specialized motion primitive-based network architecture $\pi_\phi$ that effectively estimates the optimal trajectory's mixture distribution within the workspace and further refines the corridors. Firstly, to mitigate the risk of overfitting due to absolute coordinates in the world frame, we transform both the start and end states of the trajectory planning problem into the robot's current local coordinate system. Moreover, to ensure more stable training and enhance gradient descent efficiency, we preprocess the depth values of the image by clipping and normalizing them to a range of 0 to 1, with the maximum depth value set to 10 meters. Our network begins by utilizing residual convolutions to encode multiple frames of depth images. Simultaneously, it employs multilayer perceptrons (MLPs) to encode the start and end states along

with the past odometry frames. The outputs from these processes are then seamlessly integrated to form the latent feature, denoted as $\mathcal{M}$. Assuming we have pre-constructed an offline motion primitive library $\mathbb{M}$ that represents the spatial topology, the latent feature will then be fed into another multilayer perceptron network to output a probability distribution $\mathcal{N}$ over this primitive library. It is worth mentioning that each motion primitive $\mu \in \mathbb{M}$, in order to align with the subsequent corridor parameters, is represented by $N\lambda$ points. To enhance the accuracy, the selected motion primitive $\mu^*$ is further coupled with the latent features $\mathcal{M}$ and fed into the final corridor refinement layer. This layer applies the precise positional adjustment $\Delta \boldsymbol{p}$ to each point $\boldsymbol{p}$ on the motion primitive, treating the adjusted result as the corresponding sphere center. Additionally, this module is responsible for assigning the corresponding safety radius to each sphere. This primitive-based network architecture essentially approximates the spatial mixture distribution of the optimal trajectory, thereby matching the inherently multimodal nature of local planning problems. Furthermore, this unique network structure has the potential to endow our planner with the capability to explore multiple topological spaces. Next, we discuss the method of constructing the motion primitive library

Assuming we have pre-collected hundreds of thousands of UAV flight trajectories, each trajectory is uniformly discretized into $N\lambda$ points $\boldsymbol{P}^{world} = [\boldsymbol{p}_1^{world}, ..., \boldsymbol{p}_{N\lambda}^{world}] \in \mathbb{R}^{N\lambda \times 3}$ and transformed into the local coordinate system of the robot:

$$\boldsymbol{p}_i^{local} = \boldsymbol{R}_b^{\mathrm{T}}(\boldsymbol{p}_i^{world} - \boldsymbol{b}), \forall i \in \{1, ..., N\lambda\}, \tag{11}$$

where $\boldsymbol{R}_b$ and $\boldsymbol{b}$ represent the rotation matrix and position offset of the robot's current body frame relative to the world coordinate system, respectively. To eliminate unnecessary duplicate motion primitives and limit the size of the library, as shown in Figure 2. B, we define another navigation frame $\boldsymbol{R}_{nav} = [\boldsymbol{x}^{nav}, \boldsymbol{y}^{nav}, \boldsymbol{z}^{nav}] \in \mathbb{R}^{3 \times 3}$ based on the body frame, with its origin coinciding with the start point of the body frame and the X-axis pointing towards the endpoint position:

$$\boldsymbol{x}^{nav} = \frac{\boldsymbol{p}_{end}^{local} - \boldsymbol{p}_{start}^{local}}{||\boldsymbol{p}_{end}^{local} - \boldsymbol{p}_{start}^{local}||_2}, \tag{12}$$

$$\boldsymbol{y}^{nav} = \frac{\boldsymbol{e}_3 \times \boldsymbol{x}^{nav}}{||\boldsymbol{e}_3 \times \boldsymbol{x}^{nav}||_2}, \tag{13}$$

$$\boldsymbol{z}^{nav} = \boldsymbol{x}^{nav} \times \boldsymbol{y}^{nav}, \tag{14}$$

where $\boldsymbol{e}_3 = [0, 0, 1]^{\mathrm{T}}$. Similar to Eq. (11), by transferring the motion primitives from the body frame to the navigation frame defined here, we achieve regularization in the direction towards the endpoint. Furthermore, to avoid a large number of redundant motion primitives that are actually very similar in shape but differ in spatial scale during planning, we also normalize the distance to the endpoint. The resulting processed motion primitives $\boldsymbol{P}^{normalized}$ are as follows:

$$\boldsymbol{p}_i^{normalized} = \frac{\boldsymbol{R}_{nav}^{\mathrm{T}}(\boldsymbol{p}_i^{local} - \boldsymbol{p}_{start}^{local})}{||\boldsymbol{p}_{end}^{local} - \boldsymbol{p}_{start}^{local}||_2}, \forall i \in \{1, ..., N\lambda\}. \tag{15}$$

Finally, we employ the K-Means (MacQueen et al., 1967) algorithm to cluster the processed dataset and collect 30 elite motion primitives as the library.

### 4.3 DIFFERENTIABLE TRAJECTORY OPTIMIZATION LAYER

Before discussing gradient backpropagation, we first reformulate the inner optimization in Eq. (8) and introduce its solution approach (forward process). Similar to the approach (Zhou et al., 2022), we relax the original time-continuous constraint Eq. (2,3) using a time-discrete penalty term. Consequently, the original trajectory planning problem is reformulated as an unconstrained nonlinear optimization:

$$\min_{\xi(\boldsymbol{Q}, T)} L_{\mathcal{F}(\zeta_\phi, \gamma_\phi)} = \int_0^T (\xi^{(u)}(t))^{\mathrm{T}} \boldsymbol{W} \xi^{(u)}(t) dt + \rho T$$

$$+ \sum_{i=1}^N \sum_{j=1}^\lambda w_{\mathcal{C}} \mathrm{L}_1(\mathcal{C}(\xi_{i,j}, \xi_{i,j}^{(1)}, ..., \xi_{i,j}^{(u)})) + w_{\mathcal{F}} \mathrm{L}_1(||\xi_{i,j} - \zeta_{\phi,i,j}||_2^2 - \gamma_{\phi,i,j}^2). \tag{16}$$

Here, $w_{\mathcal{C}}$ and $w_{\mathcal{F}}$ are the weights corresponding to the penalty terms. The physical meaning of $||\xi_{i,j} - \zeta_{\phi,i,j}||_2^2 - \gamma_{\phi,i,j}^2$ is the violation of trajectory points within the spatial corridor constraints,

denoted as $\mathcal{S}_{i,j}$ for simplicity. $\mathrm{L}_1(\cdot)$ is a first-order relaxation function to guarantee the continuous differentiability and non-negativity of penalty terms:

$$\mathrm{L}_1(x) = \begin{cases} 0 & x \leq 0, \\ -\dfrac{1}{2a_0^3}x^4 + \dfrac{1}{a_0^2}x^3 & 0 < x \leq a_0 \\ x - \dfrac{a_0}{2} & a_0 < x. \end{cases} \tag{17}$$

Here $a_0 = 10^{-4}$ is the demarcation point. This reformulated problem can then be robustly solved using common gradient-based numerical solvers, such as L-BFGS (Liu & Nocedal, 1989).

Defining $\xi^*$ is the optimal solution to this optimization problem Eq. (16), and $\mathcal{L}_\xi$ is the evaluation loss applied to the trajectory during training, the gradient of the neural network can be computed as follows:

$$\nabla_\phi \mathcal{L}_\xi = \nabla_\phi \mathcal{F} \nabla_\mathcal{F} \xi^* \nabla_{\xi^*} \mathcal{L}_\xi, \tag{18}$$

where all gradient derivations in the paper adhere to the denominator layout[1]. Generally, the term $\nabla_{\xi^*} \mathcal{L}_\xi$ can be analytically computed and $\nabla_\phi \mathcal{F}$ is also easily computed using automatic differentiation based on the network structure. Therefore, we primarily focus on discussing the computation of the gradient of the optimal solution with respect to the spatial constraint descriptors $\nabla_\mathcal{F} \xi^*$. Due to the use of gradient-based numerical solvers, an intuitive method for estimating parameter gradients, known as unrolling Finn et al. (2017); Bhardwaj et al. (2020); Pearlmutter & Siskind (2008); Zhang & Lesser (2010); Han et al. (2017), involves maintaining the entire computational graph throughout the iteration process. However, this approach presents significant challenges in terms of memory usage and efficiency, particularly when dealing with complex problem formulations. Additionally, it may also encounter issues related to gradient divergence or vanishing (Pineda et al., 2022). In this work, since we have already obtained the optimal solution $\xi^*$ to the problem, we use the implicit function differentiation theorem (Dontchev & Rockafellar, 2009) to analytically derive the gradients without the need for explicit unrolling of the entire iteration process. Based on the first-order optimality condition of nonlinear programming (Amos & Kolter, 2017), the optimal solution of Eq. (16) should satisfy the following equation:

$$\nabla_\xi L_{\mathcal{F}(\zeta_\phi, \gamma_\phi)}(\xi^*) = \nabla_\xi J(\xi^*) + \sum_{i=1}^N \sum_{j=1}^\lambda w_\mathcal{C} \mathrm{L}_1'(\mathcal{C}_{i,j}^*) \sum_{k=1}^u \nabla_\xi \xi_{i,j}^{(k)}(\xi^*) \nabla_{\xi_{i,j}^{(k)}} \mathcal{C}(\xi_{i,j}^{(k)*})$$

$$+ 2w_\mathcal{F} \mathrm{L}_1'(\mathcal{S}_{i,j}^*) \nabla_\xi \xi_{i,j}(\xi^*)(\xi_{i,j}^* - \zeta_{\phi,i,j}) = 0. \tag{19}$$

Here, as an example, $\nabla_\xi \xi_{i,j}^{(k)}(\xi^*)$ represents the gradient of the higher-order state of the constraint point $\xi_{i,j}^{(k)}$ with respect to the trajectory $\xi$ when $\xi = \xi^*$. Then, we apply the total differential operator to this first-order optimality condition Eq. (19):

$$(\nabla_{\xi,\xi} L_{\mathcal{F}(\zeta_\phi,\gamma_\phi)}(\xi^*))^\mathrm{T} d\xi^* + (\nabla_{\xi,\zeta_\phi} L_{\mathcal{F}(\zeta_\phi,\gamma_\phi)}(\xi^*))^\mathrm{T} d\zeta_\phi + (\nabla_{\xi,\gamma_\phi} L_{\mathcal{F}(\gamma_\phi,\gamma_\phi)}(\xi^*))^\mathrm{T} d\gamma_\phi = 0, \tag{20}$$

where each term here can be analytically derived, such as the partial differential formula for the network output shown below:

$$\nabla_{\xi,\zeta_\phi} L_{\mathcal{F}(\zeta_\phi,\gamma_\phi)}(\xi^*) = -2w_\mathcal{F} \sum_{i=1}^N \sum_{j=1}^\lambda (2\mathrm{L}_1''(\mathcal{S}_{i,j}^*)\mathcal{D}_{i,j}\mathcal{D}_{i,j}^\mathrm{T} + \mathrm{L}_1'(\mathcal{S}_{i,j}^*)\boldsymbol{I})(\nabla_\xi \xi_{i,j}(\xi^*))^\mathrm{T}, \tag{21}$$

$$\nabla_{\xi,\gamma_\phi} L_{\mathcal{F}(\gamma_\phi,\gamma_\phi)}(\xi^*) = -4w_\mathcal{F} \sum_{i=1}^N \sum_{j=1}^\lambda \gamma_{\phi,i,j} \mathrm{L}_1''(\mathcal{S}_{i,j}^*)\mathcal{D}_{i,j}^\mathrm{T}(\nabla_\xi \xi_{i,j}(\xi^*))^\mathrm{T}, \tag{22}$$

where $\mathcal{D}_{i,j} = \xi_{i,j}^* - \zeta_{\phi,i,j}$. Subsequently, through matrix operations, we equivalently transform Eq. (20) into the following compact form:

$$d\xi^* = (\nabla_{\xi,\xi} L_{\mathcal{F}(\zeta_\phi,\gamma_\phi)}(\xi^*))^{-1} \begin{bmatrix} \nabla_{\xi,\zeta_\phi} L_{\mathcal{F}(\zeta_\phi,\gamma_\phi)}(\xi^*) \\ \nabla_{\xi,\gamma_\phi} L_{\mathcal{F}(\gamma_\phi,\gamma_\phi)}(\xi^*) \end{bmatrix}^\mathrm{T} \begin{bmatrix} d\zeta_\phi \\ d\gamma_\phi \end{bmatrix} \tag{23}$$

By solving this system of equations, we can analytically obtain the desired Jacobian matrix $\nabla_{\mathcal{F}(\zeta_\phi,\gamma_\phi)} \xi^*$ in Eq. (18), which in turn allows us to derive the final parameter gradients $\nabla_\phi \mathcal{L}_\xi$:

$$\nabla_\phi \mathcal{L}_\xi = \nabla_\phi \mathcal{F} \begin{bmatrix} \nabla_{\xi,\zeta_\phi} L_{\mathcal{F}(\zeta_\phi,\gamma_\phi)}(\xi^*) \\ \nabla_{\xi,\gamma_\phi} L_{\mathcal{F}(\gamma_\phi,\gamma_\phi)}(\xi^*) \end{bmatrix} (\nabla_{\xi,\xi} L_{\mathcal{F}(\zeta_\phi,\gamma_\phi)}(\xi^*))^{-1} \nabla_{\xi^*} \mathcal{L}_\xi. \tag{24}$$

---

[1] https://en.wikipedia.org/wiki/Matrix_calculus

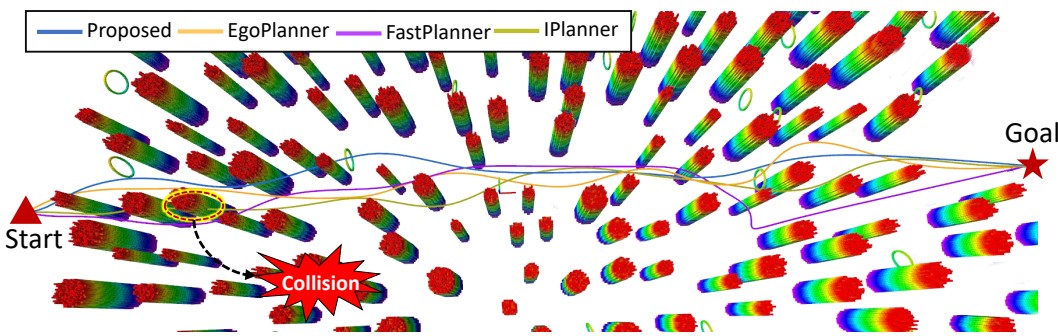

Figure 3: The simulation scenario and trajectory visualization at high aggressiveness.

## 5 EVALUATIONS

We execute all training procedures on an Nvidia RTX 4090 GPU with a batch size of 64, utilizing the Adam optimizer with a learning rate of 1.0e-5. Details on the training procedure and loss functions can be found in the Appendix B. Subsequently, all testing within the simulation environment is conducted on an RTX 2060 GPU, an Intel 10700 CPU, and an Ubuntu 20.04 operating system. For real-world experiments, we deploy them on a fully autonomous quadcopter equipped with a RealSense D430 camera and NVIDIA Jetson Orin NX. The detailed hardware and software specifications are outlined in the Appendix C.

### 5.1 BENCHMARKS

In this section, we compare our approach with two traditional algorithms and a SOTA learning-based method that receive widespread acclaim and is applied in real-world environments: **1. Fast-Planner** (Zhou et al., 2019): A hierarchical motion planner where kinodynamic search is used to construct an initial solution, followed by using ESDF to enforce safety constraints required for subsequent trajectory optimization. **2. Ego-Planner** (Zhou et al., 2020): A lightweight optimization-based motion planner that utilizes graph search to obtain a collision-free guiding path, guiding trajectories to safe regions, thus eliminating the need for ESDF construction. **3. IPlanner** (Yang et al., 2023): A powerful learning-based motion planner integrates the core concept of implicit optimization (IO) into the overarching pipeline of High-Speed Flight (HSF) (Loquercio et al., 2021), demonstrating enhanced performance in experiments. Utilizing IO, Iplanner reduces the dependency on expert trajectory data compared to HSF, and also possesses enhanced generalization performance. Moreover, by directly approximating the original optimization problem, it exhibits a better performance in terms of optimality and constraint satisfaction compared to HSF. For comprehensive comparison, we categorize scenarios into low, medium, and high aggressiveness based on varying velocity and acceleration limits of the robot. For each scenario, we randomly generate 200 navigation tasks, each approximately 70 meters in distance, within previously unseen environments containing hundreds of obstacles, as illustrated in Figure 3. Moreover, various custom parameters of traditional methods, such as convergence accuracy and the number of iterations for each case, are fine-tuned by experienced researchers in the field. For each planning instance, the local target is selected from the straight line pointing towards the destination at a specific distance (12m) from the current position. Furthermore, the replanning logic for each planner follows a time-triggered approach, with a fixed replanning frequency of 10Hz. During each replanning process, we output the three most probable candidate corridors and optimize trajectories in parallel. The execution trajectory is selected based on a weighted combination of trajectory quality $J(\xi)$ and corridor probability $P(\mathcal{F})$: $J(\xi) + w(1 - P(\mathcal{F}))$, with a weight $w$ of 400.

We collect and summarize the dynamic parameters and success rate during flight, presenting the results in Table 1. In various settings, our method demonstrates the highest success rate. Although the Ego-planner also shows impressive robustness at low aggressiveness, its success rate drops sharply in high aggressiveness scenarios. This is because the Ego-planner uses an iterative framework for obstacle avoidance that lacks convergence guarantees, often failing to find safe solutions in challenging high-speed cases. Moreover, in terms of optimality, our method achieves a shorter flight time and effectively reduces peak jerk compared to the Ego-planner, which means our flights are smoother and place less strain on the motors, thereby better preserving hardware longevity (Mellinger & Kumar, 2011a). As for the Fast-planner, it requires maintaining an ESDF, which significantly increases mapping time, resulting in planning delays that are tens of times longer than our method. This delay reduces the system's responsiveness to unknown environments, particularly affecting performance at high speeds. Moreover, the substantial computational power required for mapping on onboard computers limits the resources available for other modules, such as localization, thereby compromising the overall system robustness. Additionally, the discrete nature of the search strategy in this method lacks completeness within limited computation time, further reducing success rate. Although the learning-based Iplanner demonstrates

Table 1: **Quantitative benchmarks in various cases.** "Delay Time" refers to the computational time required by each module after receiving the depth image. For the ego planner, "Mapping" is the time to transform the depth image into a grid map via raycasting. "Solution" is the time spent on trajectory optimization. In Fastplanner, "Mapping " additionally includes the maintenance of ESDF, and "Solution" also additionally encompasses the time spent utilizing kinodynamic search to acquire initial values required for optimization. For our approach, "Model" denotes neural network inference time, and "Solution" is the subsequent numerical optimization time. In assessing dynamic metrics, we capture the maximum velocity (M.V), acceleration (M.A), and jerk (M.JK) in each navigation mission and subsequently calculate the average and peak values derived from these metrics across multiple navigations. For instance, in the case of M.V, the external value signifies the average of the maximum velocities recorded in multiple navigations, whereas the internal value illustrates the absolute peak value. "Execution Time" is the average flight duration per navigation task. A task is a failure if the robot contacts any obstacle; otherwise, it is successful. Constraint violations are marked in red, while the highest success rates are highlighted in blue.

| Methods | Delay Time ($ms$) | | | | Dynamics | | | Execution | Success |
|---|---|---|---|---|---|---|---|---|---|
| | Mapping | Model | Solution | Total ↓ | M.V $(m/s)$ | M.A $(m/s^2)$ | M.JK ↓ $(m/s^3)$ | Time ↓ $(s)$ | Rate ↑ $(\%)$ |
| Low Aggressiveness $v_{limit} = 2m/s, a_{limit} = 3m/s^2$ | | | | | | | | | |
| Proposed | 0.0 | 4.1 | 1.3 | 5.4 | 2.0 (2.0) | 1.5 (2.0) | **6.9 (13.6)** | **35.45** | **100.0** |
| Fastplanner | 54.9 | 0.0 | 6.0 | 60.9 | 2.0 (2.0) | 1.8 (2.9) | 9.4 (32.1) | 36.33 | 92.5 |
| EgoPlanner | 5.5 | 0.0 | 1.8 | 7.3 | 2.0 (2.0) | 2.2 (3.0) | 15.7 (27.6) | **35.45** | **100.0** |
| Iplanner | 0.0 | 3.0 | 0.0 | **3.0** | 2.3 (5.4) | 3.4 (14.8) | 23.9 (124.6) | 42.67 | 93.0 |
| Medium Aggressiveness $v_{limit} = 5m/s, a_{limit} = 6m/s^2$ | | | | | | | | | |
| Proposed | 0.0 | 4.0 | 1.2 | 5.2 | 5.0 (5.0) | 6.0 (6.0) | **29.5 (50.0)** | 13.66 | **97.5** |
| Fastplanner | 60.0 | 0.0 | 4.9 | 64.9 | 5.0 (5.0) | 5.9 (6.0) | 30.9 (**46.1**) | **13.54** | 88.0 |
| EgoPlanner | 6.7 | 0.0 | 3.0 | 9.7 | 5.0 (5.0) | 6.0 (6.0) | 50.1 (89.2) | 14.67 | 91.0 |
| Iplanner | 0.0 | 3.0 | 0.0 | **3.0** | 4.8 (4.9) | 6.5 (22.5) | 41.4 (362.7) | 13.69 | 90.5 |
| High Aggressiveness $v_{limit} = 8m/s, a_{limit} = 10m/s^2$ | | | | | | | | | |
| Proposed | 0.0 | 4.0 | 0.9 | 4.9 | 8.0 (8.0) | 6.5 (8.5) | **27.5 (54.4)** | 10.13 | **93.0** |
| Fastplanner | 62.3 | 0.0 | 3.9 | 66.2 | 7.9 (8.0) | 7.52 (10.0) | 39.4 (71.5) | 10.50 | 75.0 |
| EgoPlanner | 6.7 | 0.0 | 5.5 | 12.2 | 7.5 (8.0) | 9.5 (10.0) | 118.2 (311.9) | 12.88 | 78.0 |
| Iplanner | 0.0 | 3.1 | 0.0 | **3.1** | 7.6 (8.0) | 6.8 (44.3) | 51.6 (629.5) | **9.28** | 71.5 |

the highest time efficiency, our method is only $2ms$ slower, which is also greatly satisfies the real-time require-ments of the system. However, IPlanner's lack of consideration for multimodal problem characteristics makes it prone to unsafe local optima. Notably, although Iplanner incorporates penalties for dynamic constraints dur-ing offline training based on IO to guide the network towards constraint compliance, it is less comprehensive compared to our explicit spatial-temporal optimization applied directly to the trajectory online. For instance, In highly aggressive scenarios, the peak acceleration exceed constraints by more than four times, which is clearly outside the robot's expected operational mode and risks severe crashes.

## 5.2 REAL-WORLD EXPERIMENTS

To validate the effectiveness of our framework in the real world, we create a point cloud simulating a wooded area and utilize this environment to generate the dataset and train the network. In the simulation, the depth images are obtained just by ray casting in parallel using GPU, without further processing the depth data to make it close to that in the real world as Song et al. (2020). Instead, we chose to patch the depth images from

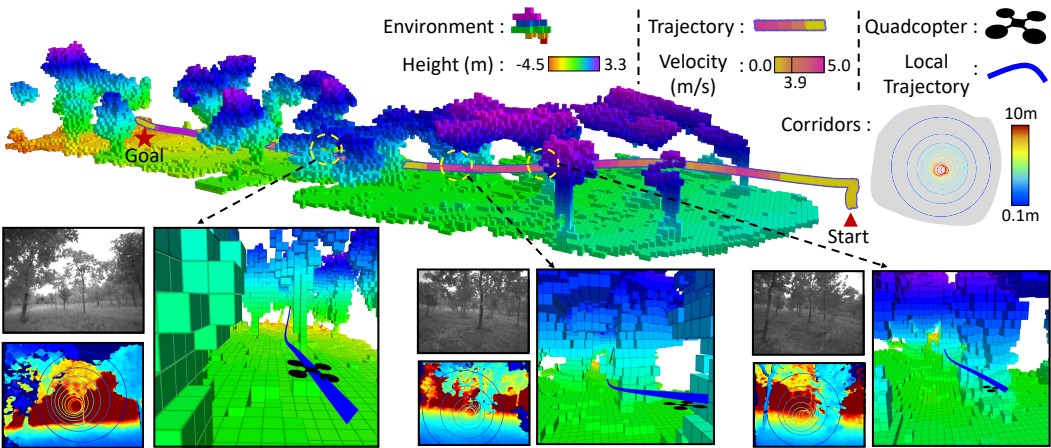

Figure 4: Real-world Experiments.

Table 2: **Statistics in Real-World Experiments.** Speed and attitude data of the drone are derived from visual-inertial odometry. Acceleration is the norm of the acceleration vector in the world frame minus gravity. We use the attitude of the drone to transfer the data from the accelerometer in the IMU to the world frame. The angular velocity data comes from the gyroscope of the IMU, which is in the body frame.

| Statistic | Speed (m/s) | Roll (deg) | Pitch (deg) | Yaw (deg) | Acceleration (m/s$^2$) | Norm of Angular Velocity (rad/s) |
|---|---|---|---|---|---|---|
| Max | **5.0** | 6.9 | 25 | 9.3 | **5.6** | 2.7 |
| Mean | 3.7 | 0.068 | 5.7 | -0.49 | 1.6 | 0.47 |
| STD | 1.2 | 2.6 | 10 | 3.9 | 1.2 | 0.44 |

the D430 in the real world to reduce the sim2real gap, which would even allow the drone to fly agilely when the camera lens are blurred, as demonstrated in Appendix C.4.

We conduct real-world experiments in a wooded area, randomly selecting goals more than 50m away from the robot, one of which is illustrated in Figure 4. In this case, the maximum speed and acceleration of drone are limited to $5m/s$ and $6m/s^2$ respectively. The grid map representing the environment is obtained by recording rosbag and projecting the depth data to the world frame by aligning the depth images and odometry after the experiment.

Table 2 shows statistics in the real-world experiments corresponding to the case shown in Figure 4. As we can see, the robot maintains a relatively high speed throughout to reach the target state without violating the maximum speed and acceleration constraints we set. Curves of some physical quantities and more moments from the real-world experiments are detailed in Appendix C.3.

## 6 CONCLUSION

In this paper, we propose an end-to-end visual navigation system that learns optimal trajectories from depth images and integrates a spatial-temporal trajectory optimizer to ensure adherence to kinematic constraints, enhancing interpretability and debugging. Additionally, our regularized trajectory design ensures thorough exploration of feasible spaces with minimal memory consumption. Extensive experiments demonstrate that our method excels in guiding drones through complex environments, outperforming both traditional and learning-based methods. Real-world tests on an onboard computer confirm the system's capability for high-speed, safe flight in dense forests. In the future, we plan to expand our method by incorporating task-specific objective functions or constraints into the optimization process to better adapt to specific task scenarios, such as exploration and tracking. Moreover, we intend to imbue the extracted spatial corridors with temporal information to enhance UAV robustness against dynamic obstacles and enable application in multi-robot collaboration.

## 7 REPRODUCIBILITY STATEMENT

To ensure the reproducibility of our work, we provide the following illustrations in appendix and supplementary material:

- **Training Setup**: We provide details of the training in Appendix B, including how to collect data and configure the training strategy.
- **Simulation Environment**: We provide the code and guidelines used to deploy the proposed framework in a simulation environment based on Robot Operating System (ROS) in the supplementary material.
- **Real Robot Setup**: In Appendix C, we provide the details of the real-world experiments, including the software and hardware we use.

Besides, to make it even easier for interested researchers to reproduce our work, we promise to further open source more complete code at a later date, encompassing everything from data collection to deployment, in a github repository with an easy-to-follow guide. We believe the open source repository will be an important contribution to the community.

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

## A  OPTIMALITY ANALYSIS

In our work, we aim to solve the decomposed dual-layer optimization problem Eq. (6-8). However, we argue that we do not compromise on optimality because this decomposition is tight, implying that the dual-layer optimization and the original optimization problem Eq. (1-5) share the same optimal solution. In other words, for the optimal solution of the original problem $\xi^*, \mathcal{F}^*$, it necessarily satisfy the following conditions:

$$\xi^* = \arg\min_{\xi} J = J(\xi), s.t.Eq.(2-5)(\mathcal{F}^*). \tag{25}$$

Here we give the proof by contrapositive. If the optimal solution of the original problem is not the optimal solution of the subproblem, let's assume that the optimal solution of the subproblem is represented by $\hat{\xi}$, where $J(\hat{\xi}) < J(\xi^*)$. Thus, we obtain a new set of solutions, $\hat{\xi}$ and $\mathcal{F}^*$, which necessarily satisfy the constraints of the original problem and have a more optimal objective function than the original solution $\xi^*, \mathcal{F}^*$. That is, for the original problem, we have a set of better solutions $\hat{\xi}, \mathcal{F}^*$, which contradicts the assumption that $\xi^*, \mathcal{F}^*$ are the optimal solution of the original problem. Therefore, the hypothesis is not valid, and the original proposition is proven.

## B  IMPLEMENTATION DETAILS

### B.1  DATA COLLECTION

In our experiments, we conduct training across various parameter settings. For each configuration, we randomly generate 200 forest environments, each measuring 75m by 75m, and sample numerous start and end points to construct motion planning problems. To collect trajectory and flight sensor data, we first use high-resolution hybrid A* as the frontend method in the simulation environment to search for a rough topological path. This path then serves as an initial guess for further backend trajectory optimization. Additionally, for obstacle avoidance, we maintain a precise ESDF to push the trajectory away from obstacles. The local ESDF is set to cover 14 meters forward and backward in the local coordinate system, encompassing the entire local trajectory (around 12 meters long) to prevent map boundary issues during optimization and enhance trajectory foresight. However, maintaining a large ESDF field and the delays between perception, frontend, and backend processes result in a total navigation framework latency of a few hundred milliseconds, even on high-performance personal computers, making it unsuitable for high-speed UAV flight scenarios. Therefore, to successfully collect data in the simulation environment, we deliberately adjust the simulator's time to 1/10 of real-time. Although this method reduces data collection efficiency, the server's excellent multi-threading capabilities allow us to collect data in large batches. Even with a target of 400000 data points, the total collection time does not exceed 10 hours. It is worth noting that due to perception errors and various infeasible local optimizations, the collected local trajectories may not always be completely safe or physically feasible. However, through our designed unsupervised loss functions and gradually transitioning to a reference-free training strategy, we reduce our method's dependence on reference trajectories and enhance robustness.

### B.2  LOSS FUNCTION

Our network architecture is based on following a trajectory library, which approximates a spatial mixture distribution of near-optimal trajectories. Consequently, the evaluation function of the network is designed as the expected loss of the policy: $\overline{\mathcal{L}}_\phi = \mathbb{E}_{\mathcal{F} \sim P_\phi(\mathcal{F})}[\mathcal{L}(\mathcal{F})] = \sum_{\mathcal{F}} \mathcal{L}(\mathcal{F})P_\phi(\mathcal{F})$. In implementation, we output the probability distribution over a discrete set of motion primitives, and each generated flight corridor deterministically depends on the selected primitive without any randomness: $P_\phi(\mathcal{F}) = P_\phi(\mathcal{F}(\mu)) = P_\phi(\mu)$. Consequently, the expected loss of the network is instantiated as follows:

$$\overline{\mathcal{L}}_\phi = \sum_{\mu \in \mathbb{M}} P_\phi(\mu)(w_J J(\xi^*(\mathcal{F}(\mu)) + w_{sf}\mathcal{L}_{safe}(\mathcal{F}(\mu)) + w_{fb}\mathcal{L}_{feasible}(\mathcal{F}(\mu)))). \tag{26}$$

Here, $w_J = 0.05, w_{sf} = 10$ and $w_{fb} = 0.05$ are the weights corresponding to the respective losses. $P_\phi(\mu)$ represents the probability of selecting motion primitive $\mu$, which is output by the network. $J$ is the evaluation metric of the trajectory optimized based on the flight corridor $\mathcal{F}$, defined as a combination of energy and execution time, as described in Eq. (1). $\mathcal{L}_{safe}$ represents the safety condition of the corridor, acting as a penalty term during training to push the corridor away from obstacle regions. To prevent the generation of unreasonable safety corridors by the network's initial parameters, which could render the inner optimization problem infeasible, we introduce $\mathcal{L}_{feasible}$ to penalize such infeasible corridors.

Due to our use of a series of spheres $(\zeta, \gamma)$ to represent $\mathcal{F}$, the safety condition is modeled as follows:

$$\mathcal{L}_{safe}(\mathcal{F}) = \sum_{i=1}^{N} \sum_{j=1}^{\lambda} \mathrm{L}_1(\gamma_{i,j} - \mathcal{S}_d(\zeta_{i,j}, \epsilon)), \tag{27}$$

where $\mathcal{S}_d$ is the signed distance from the sphere center to the obstacle, which can be efficiently obtained via trilinear interpolation from a precomputed ESDF field $\epsilon$ bound to the local environment. Furthermore, we utilize an ELU function with a constant bias as the activation function for the final layer outputting the radius, ensuring that the sphere is always larger than the robot's shape. Upon obtaining the inner optimization output, we verify the corridor constraints Eq. (3) to assess the degree of violation. If the constraint violation exceeds a predetermined threshold $\delta = 0.0005$, the output corridor is deemed invalid and a penalty is applied. To mitigate the infeasibility of the corridor, we adjust the sphere by moving its center closer to the detached constraint point $\hat{\xi}_{i,j}$ and appropriately increasing its radius:

$$\mathcal{L}_{feasible}(\mathcal{F}) = \mathrm{L}_1(-\delta + \sum_{i=1}^{N} \sum_{j=1}^{\lambda} ||\hat{\xi}_{i,j} - \zeta_{i,j}||_2^2 - \gamma_{i,j}^2). \tag{28}$$

Commencing training directly on the original task Eq. (26) could propel the network to undesirable local minima or saddle points. To counter this predicament, we employ a curriculum learning strategy, progressively guiding the network to navigate the complexity of the problem in a well-structured, incremental manner. Initially, we decouple the network's primitive probability layer from the corridor refinement layer, training each layer on their respective outputs independently. Notably, both layers share a common perceptual convolutional layer. The primitive probability layer involves the calculation of the cross-entropy loss between the output probability distribution and the ground truth labels: $-\sum_{\mu \in \mathbb{M}} y_\mu \log(P_\phi(\mu))$, where $y_\mu$ is the ground truth label. Conversely, the corridor refinement layer, during its training phase, avoids selecting the primitive with the highest probability output generated by the network. Instead, the ground truth primitive $\mu^*$ is directly input into the model. The centers of the resulting corridor $\mathcal{F}(\mu^*)$ are supervised by the constraint points of the offline-collected reference trajectory, and the radii are supervised by the signed distance to obstacles at those points. The losses from the primitive probability layer and the corridor refinement layer are combined using a weighted sum, which is then backpropagated to update the network parameters. Ultimately, after the training approaches convergence, we build upon the foundation of the previously trained model to reconnect the primitive probability layer and the corridor refinement layer, facilitating further refinement using $\overline{\mathcal{L}}_\phi$.

## C REAL-WORLD DEPLOYMENT DETAILS

### C.1 HARDWARE SETTINGS

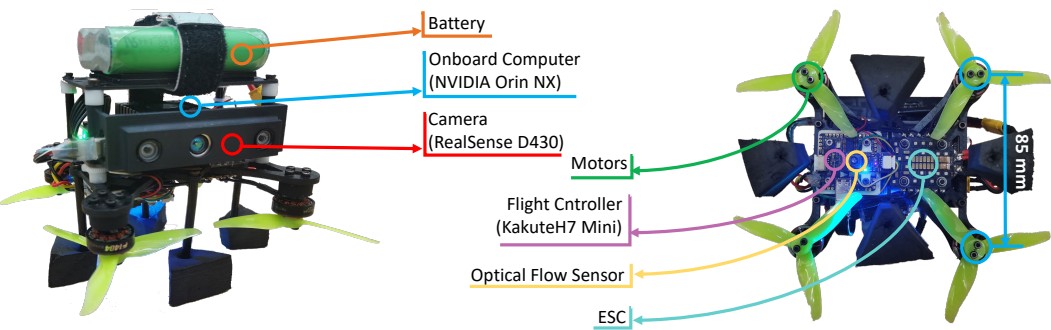

Figure 5: Hardware Settings.

All the experiments are performed on a 120-mm wheelbase microplatform that we designed and assembled. The drone is mainly made up of the following four subsystems, as shown in Figure 5.

**1) Power and movement suite**. We use a 3400-mAh LiPo battery with 12.6-V voltage as the energy source for the quadcopter. Four 4600-kv brushless motors (model 1404) incorporating 3-inch, three-blade propellers and a four-in-one electronic speed controller with a 50-A maximum current are used to constitute the power module of it. The propellers are mounted at the bottom of the airframe, and therefore, a strong downwash flow will not blow directly onto the body.

**2) Low-level control unit**. The 20 mm by 20 mm sized Holybro Kakute H7 Mini v1.3 is chosen to be the flight control unit (FCU) running PX4 Autopilot (Meier et al., 2015). The FCU is equipped with a STM32 H7 MCU and a BMI270 IMU, receiving angular velocity and thrust commands from the high-level navigation unit, controlling the pose of the UAV in $SE(3)$ space, and sending IMU data to the navigation unit.

**3) High-level navigation unit**. All the localization, planning, and high-level control codes are run in this unit, whose hardware configuration is NVIDIA Orin NX, a powerful compute for embedded and edge systems with a six-core CPU, 1024-CUDA-core GPU, and 8 GB of RAM, which is strong enough to support high-frequency calls to our algorithms.

**4) Sensors**. A grayscale and depth camera Intel Realsense D430 is used for localization and perception, where the grayscale images are fused with the IMU data from FCU and utilized by the visual-inertial odometry (VIO). The output depth images is processed and fed as the input to the planner. Besides, we use an optical flow sensor to estimate the height and correct the error of VIO.

## C.2 Software Settings

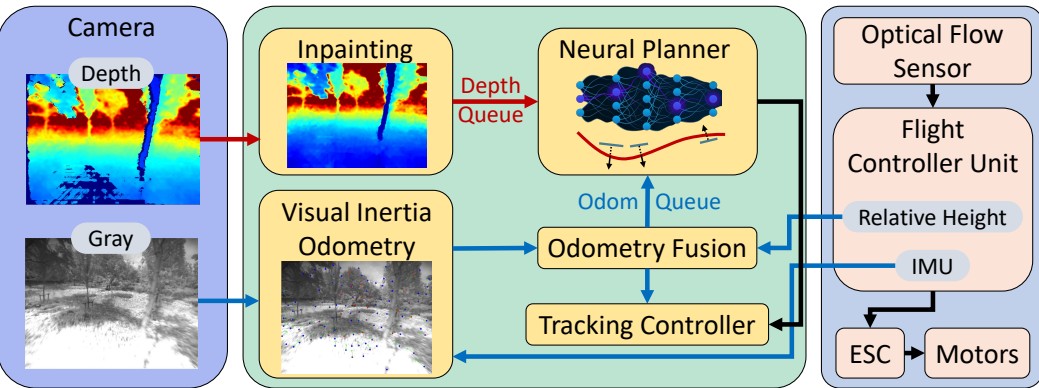

Figure 6: Software Settings.

The real-world software setup consists of three main parts, as shown in Figure 6. The grayscale images output by the D430 module and the IMU data from the FCU are received by the visual-inertial odometry module to compute the raw odometry. Based on the relative altitude values provided by the optical flow sensor, the odometry fusion module corrects the drift of the raw odometry on Z-axis of the world frame, yielding high-frequency, robust odometry for supplying to the neural planner and the trajectory tracking controller. Depth data from the D430 module pass through the image inpainting module and is synchronized with the odometry from the odometry fusion module, which are used as inputs of the neural planner. The data input to the neural planner is projected to the local frame of the UAV. After the inference of the optimization-embedded neural network, the replanned trajectory is generated at a frequency of 10Hz and is sent to the trajectory tracking controller to compute the final desired thrust and angular velocity. The obtained thrust and angular velocity commands are given to the FCU, which is converted to the final motor speed by the ESC.

**1) Multi-sensor fusion localization system**. For VIO, we refer to the framework VINS (Qin et al., 2018) and use Ceres (Agarwal et al., 2023) as the optimizer. Based on this framework, to obtain more high-frequency (200Hz) odometry for more accurate trajectory tracking control, we integrate between each odometry frame (15Hz) with the measurements of angular velocity and acceleration from the IMU. In addition, to compensate the error of VINS in Z-axis of the world frame, extended Kalman filter is adopted to fuse the altitude data from the optical flow sensor and raw odometry.

**2) Perception and motion planning**. In the real world, owing to reasons such as excessive ambient lighting, depth images from d430 are usually incomplete, with some regions of the image having no data, as shown in the upper left corner of Figure 6. To reduce the sim2real gap, treating the image as the "stream function" of a 2D incompressible flow, we propagate information into regions needing inpainting by solving an equation similar to the Navier-Stokes equations numerically (Bertalmio et al., 2001). Then, the inpainted images are downsampled to the size allowed by the neural planner, where the depth values are also limited to the range

of $0 \sim 10$m. To synchronize the depth measurements and odometry, we use synchronizer of message filter implemented in Robot Operating System (ROS) (Stanford Artificial Intelligence Laboratory et al.). Further, a finite length queue is maintained, which will be accessed when the neural planner is asked to replan. For deploying the proposed framework, we use Libtorch (Paszke et al., 2019) as the C++ library to implementing neural network inference and tensor computation. Besides, TBB (Pheatt, 2008) is adopted for multi-process management to enable parallel trajectory optimization based on multiple alternative safe corridors.

**3) Controllers** For trajectory tracking, we use a differential flatness based controller (Mellinger & Kumar, 2011b). To minimize the singularities in the attitude solution results due to the UAV approaching the singularities of the differential flatness when high maneuvering flying, we switch to quaternions to describe the attitude and adopt Hopf fibration (Watterson & Kumar, 2019) to solve the $SO(3)$ state of the UAV. We also refer to Faessler et al. (2017) to consider air resistance during high-speed flight in the differential flatness computation. After getting the desire attitude and angular velocity, we use feedforward with feedback control to calculate the final angular velocity command. In FCU, PD controller is used to control the angular velocity command, whose measurement is from filtered IMU data with Butterworth filter.

### C.3 SOME DATA FROM REAL-WORLD EXPERIMENTS

We capture some moments from real-world experiments. Figure 7 shows the behavior of the robot in the first person view while avoiding obstacles. Figure 8 shows the variation of the relevant physical quantities of the robot for one complete flight corresponding to Figure 4, where the speed and attitude data are derived from visual-inertial odometry, the angular velocity data is output by the gyroscope in the FCU. We align the attitude and body acceleration from IMU to compute the norm of the acceleration vector minus gravity.

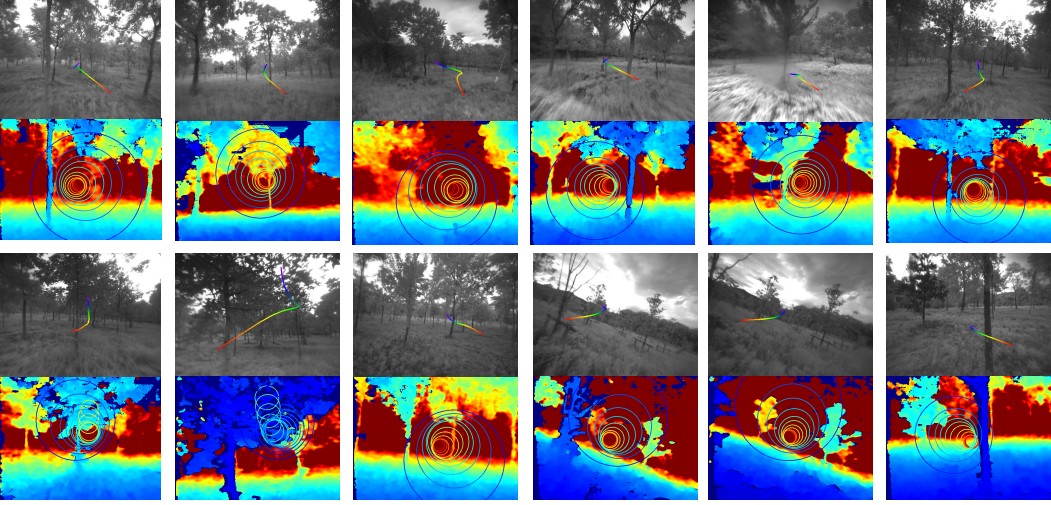

Figure 7: **Some great moments with the robot from D430 module as the first view**. We use the pose estimation of the robot from VIO and camera intrinsic matrix, extrinsic matrix to project the optimized trajectory to the gray image, represented by rainbow-colored lines, where red to purple indicate that the positions in the trajectory are near to far from the robot. In the depth images, the colors from blue to red indicate the depth values from small to large, where the circles indicate the corridor with the highest probability of the neural network output. The radius of the circle indicates the size of the corridor.

### C.4 ROBUSTNESS AND RISKY SITUATIONS

In the real world, drones can get dust, water droplets, and other foreign objects, which can contaminate the lens and thus create holes in the depth image. These untreated pixels may be perceived as obstacles by the robot and cause it to fail to pass through the region or behave conservatively. After inpainting, the robot can still able to demonstrate agile flights, as shown in Figure 9, where the grid maps in the upper right corner of two cases are constructed by aligning the odometry and inpainted depth images offline after the flight.

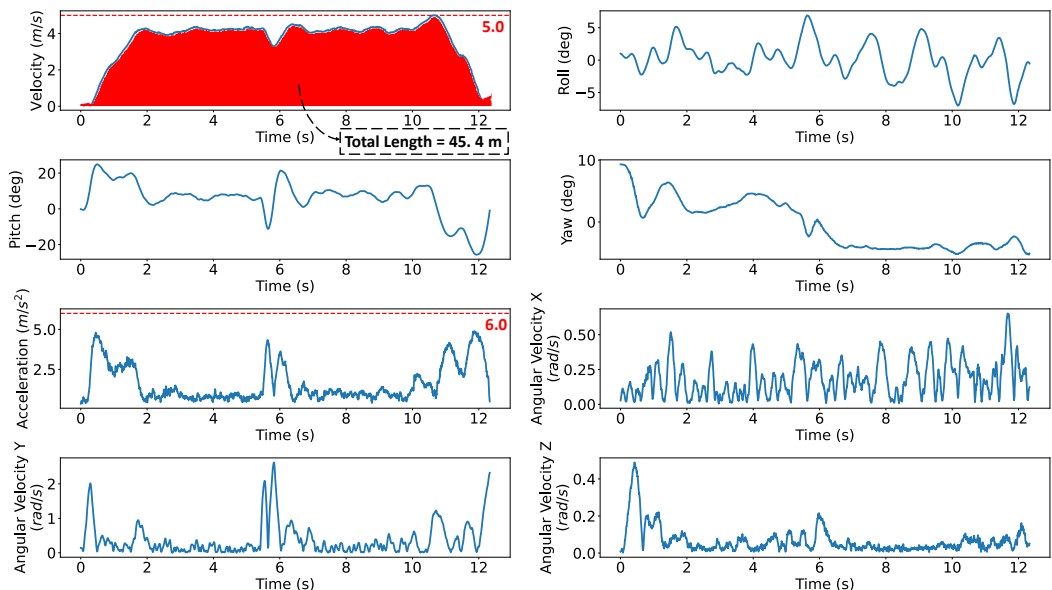

Figure 8: **Curves of some physical quantities corresponding to Figure 4**. The three axes of angular velocity are the three axes corresponding to the robot body frame, where the X-axis is perpendicular to the camera lens in the outward direction and the Z-axis is perpendicular to the paddle plane in the skyward direction. The Y-axis can then be determined via the right-hand rule.

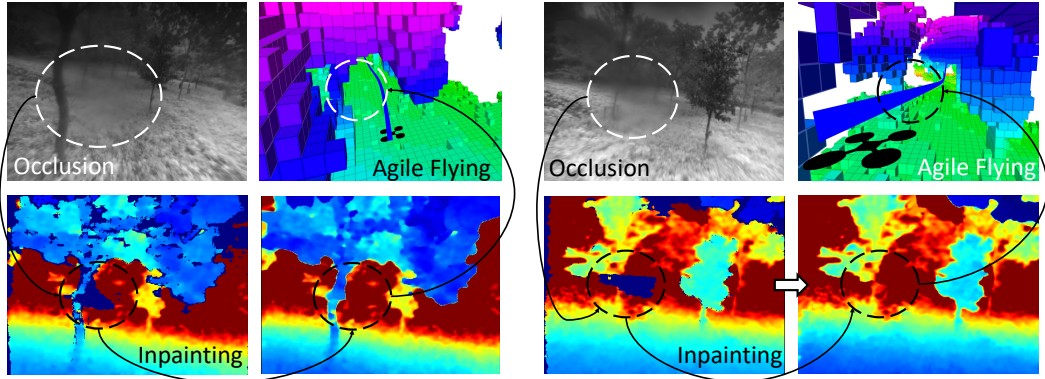

Figure 9: Image inpainting makes the system more stable.

One of the limitations of our framework is the need for the goal to be sufficiently secure. As shown in Figure 10, if the goal selected in an experiment is closer to obstacles, the robot is likely to face danger in approaching it. In the future, we will consider replacing the fixed endpoint selection strategy by utilizing another neural agent in order for the robot to intelligently choose a safe area near the goal to stop.

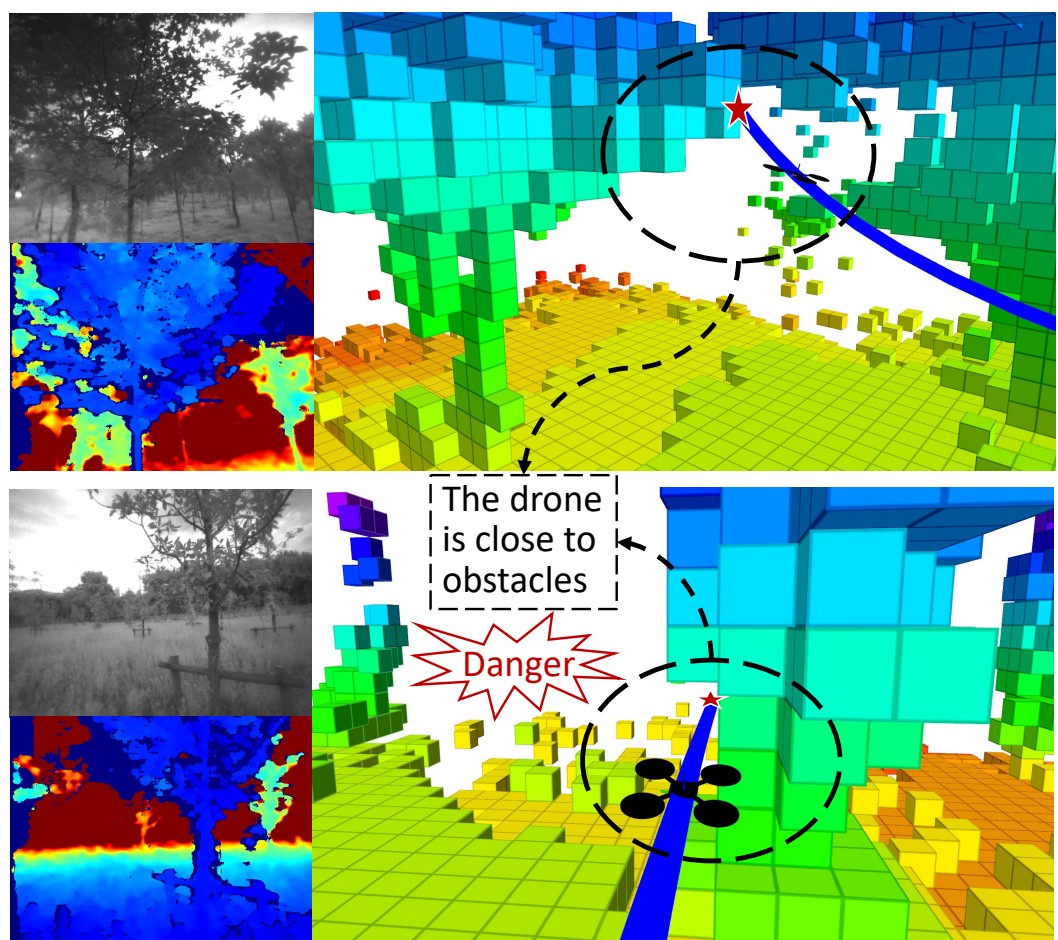

Figure 10: The drone is approaching some dangerous goals.

