# OpenReview forum: "Agile Flight  with Optimization Embedded Networks"
_ICLR.cc/2025/Conference — ICLR 2025 Conference Withdrawn Submission_

### Official Review · Reviewer_3dMp · 2024-11-03

**Soundness:** 3
**Presentation:** 3
**Contribution:** 3
**Rating:** 6
**Confidence:** 4

**Summary:**

The paper introduces a fully differentiable, end-to-end visual navigation system that generates kinodynamically feasible paths for UAVs by leveraging depth images, start and goal positions as inputs. This approach produces safe spatial corridors, which are then used for trajectory optimization, ensuring adherence to the UAV’s kinematic constraints. Through integrated training, the model effectively combines perception and planning into a cohesive system, allowing for efficient and robust path generation. Experimental results highlight the system's superior computational efficiency and effectiveness compared to existing trajectory optimization and end-to-end navigation methods.

**Strengths:**

1. The approach of generating safe corridors from depth images is compelling, as it avoids the limitations of generating potentially infeasible waypoints. By focusing on corridor generation, this method takes full advantage of trajectory optimization capabilities, resulting in more feasible and efficient paths.

2. The method’s effectiveness is demonstrated through extensive simulations and real-world experiments, underscoring its readiness for practical deployment and its robustness in real-world environments.

**Weaknesses:**

1. Unlike the previous optimization-based end-to-end system, iPlanner, which does not rely on pre-computed paths, the current method requires a library of motion primitives as training data. This dependency on pre-existing motion trajectories could limit flexibility and adaptability in environments that significantly differ from those represented in the training set.

2. Traditional trajectory optimization methods inherently provide safety guarantees by strictly adhering to hard constraints. In contrast, this approach converts constraints into soft penalties, which, while allowing for fully differentiable optimization, may risk violating critical constraints under certain conditions, potentially compromising safety.

**Questions:**

1. Motion Primitive Dependency: The end-to-end optimization in iPlanner eliminates the need for expert training data, while the current method relies on precomputed motion primitives to capture the spatial distribution of corridors. Could the generated trajectories in this method be adapted dynamically as motion primitives to enhance corridor optimization and reduce reliance on a fixed library?

2. Safety Guarantees: Traditional corridor-based methods inherently provide safety guarantees in the optimization process. In this method, however, all constraints are converted to soft penalties, which might compromise strict adherence to safety. Is there a feasible way to enforce $L_{safe}$ as a hard constraint to ensure similar safety guarantees?

3. Corridor Quality Comparison: Regarding the optimality of the generated corridors, how does the quality of the corridors produced by this method compare to those generated by search-based methods? Specifically, are the spatial coverage, tightness, and suitability for high-speed navigation improved with this approach, and are there quantitative results to support these comparisons?

---

### Official Review · Reviewer_rSkb · 2024-11-04

**Soundness:** 1
**Presentation:** 1
**Contribution:** 1
**Rating:** 3
**Confidence:** 4

**Summary:**

In robotics, agile flight with quadcopters has been an important topic for few groups of researchers. Despite application is not clear, pushing the state of the art in vision based agile flight was seen as an interesting research direction, connecting perception and action. This paper continues that research endeavor and presents a method that combines classical planning methods with learning. The key idea is to use an optimizer which is differentiable, and can be plugged in with learning methods. Such methods have been existing in robotic literature, and the current paper uses that idea. Experiments are conducted to show improvements in metrics, and one real world test with robots.

**Strengths:**

-	The direction of this research is interesting. Combining classical planners with learning based method is a trendy topic in robotic planning.

**Weaknesses:**

-	Unfortunately exposition can be improved in several ways.

Contribution should be exactly stated in the introduction. Related work should discuss similar methods that also combines classical planner and learning based planner with differentiability as a common ground. Figure 2 is difficult to parse.

-	It is unclear if ICLR is the right venue for this research since there isn’t much machine learning component apart from using neural networks.

I see more fit to ICRA or IROS for this paper, where many researchers working on similar ideas usually publish their results.

**Questions:**

Was the proposed method deployed and tested on a real robot? If so, attaching a video might be useful. Otherwise, section on real-world experiment might be misleading.

---

### Official Review · Reviewer_AEnx · 2024-11-04

**Soundness:** 3
**Presentation:** 3
**Contribution:** 3
**Rating:** 6
**Confidence:** 4

**Summary:**

This paper proposes a method that embeds optimization into the neural networks to directly generate high-quality, dynamically feasible trajectories from visual depth inputs. The authors investigate about traditional planning methods and recent learning-based methods, where traditional methods require modular decomposition which introduces physical delays, while learning-based methods can not handle the constraints on the trajectory. The authors combine the strengths of both frameworks, using training data of traditional map-based methods to train the network to avoid using map in the applications. The core contribution is innovatively incorporating the optimization module into the network and the cost function of the trajectory optimization as a part of the loss function, achieving good performance. After that, the authors conduct several simulation and real-world experiments to validate the effectiveness of the proposed method.

**Strengths:**

The biggest strength is the authors innovatively incorporate the optimization module into the network and the cost function of trajectory optimization as a part of the loss function, combining the strengths of traditional planning frameworks and learning-based methods to achieve good performance.

**Weaknesses:**

Several details are not adequately explained, where I am pointing out in the Questions below.

**Questions:**

1)In the Abstract, line 019, this sentence should be like “Neural networks establish flight corridors to serve as the spatial constraints………”

2)Line 037, “environment,explicitly”, here lacks a space.

3)Line 122, the problem of the investigated method is it can not run in “real-time” due to the offline convex decomposition. The authors here should add this.

4)Line 273, “probability distribution over this primitive library” should be explained in more detail. Such as “the probability of each trajectory in the primitive library”.

5)Line 783, the ground truth label is y_{/miu}, but I didn’t figure out how this ground label comes out. The authors use the trajectories from map-based traditional planners as data which is deterministic, but the output of the network is the probability distribution which is multi-modal and not deterministic, so the authors should add more details about how to deal with the deterministic trajectories from collected data into the probability distribution.

6)Line 778, “Commencing training directly on the original task Eq. (26) could propel the network to undesirable local minima or saddle points.” Here I don’t understand why and I need the authors to explain in more detail.

---

### Official Review · Reviewer_kde4 · 2024-11-08

**Soundness:** 3
**Presentation:** 2
**Contribution:** 2
**Rating:** 3
**Confidence:** 4

**Summary:**

The authors propose an end-to-end depth map paradigm for agile collision avoidance in the context of quadrotor flight. The approach relies on coupled training of the output trajectory mixture distribution over pre-identified primitives and the embedded trajectory optimization differentiable layer. The latter responds to input data to change the geometric space constraints (safe corridor as sequence of spheres). Demonstrations of performance in simulation as well as in the real world are provided to back the authors' claims.

**Strengths:**

- The solution is well engineered to the problem defined
     - Lightweight compute for real time inference on Orin NX
     - Efficient abstraction and simplification with few motion primitives and simple safe corridor geometry and sequencing
     - Integration of kinematic and safety constraints via an energy optimization loop running inside the neural policy
- Good visualization and figures for the work
- Ample benchmarking against baselines with solid success rate and jerk improvements while maintaining competitive maximum velocities

**Weaknesses:**

- The solution is very cutely engineered to the highly specific problem defined
    - A lot of effort is put into making the solution efficient, with limited novelty in the techniques/ model architecture used
- The authors claim that "Compared to conventional learning-based motion planning algorithms, our approach is distinguished
by the use of implicit differentiation to embed the trajectory optimization within the neural
network, enabling coupled training."
    - I would argue that this approach is far from being novel, one work that comes to mind is W. Xiao et al., "BarrierNet: Differentiable Control Barrier Functions for Learning of Safe Robot Control," in IEEE Transactions on Robotics, vol. 39, no. 3, pp. 2289-2307, June 2023,  where CBF QPs are solved during inference (with coupled training for tuning their parameters) also in the context of visual end-to-end navigation.
    - I cannot help but feel that what would be the main technical learning contribution of the paper has in fact been well explored (in forms more or less similar) for a couple of years now (see citations within/of cited work above)
- The real world environment is not cluttered enough to impress, especially given the central claim of agility

This is a really good systems/robotics paper but a mediocre ML/AI contribution.
In that respect the technical novelty as well as suitability for the venue is put into question.

**Questions:**

- What happens when N the number of pieces of the trajectory is small or if the spheres do not intersect? Does it matter that they intersect at all (optimization formulation doesn't seem to check this at all although authors claim at line 247 that "By imposing constraints at these discrete points, we can effectively control the
entire trajectory")?
    - Along the same lines, give that the corridor is based on this discretization, equation (3) doesn't guarantee collision avoidance over the entire trajectory. What ways would you see forward to solve this?
    - Is this behind the collisions accounting for the scores in the more aggressive settings or are there any other failure mode reasons?
    - This lacks a study of the effect of increasing N on the model complexity/optimization time, as well as on performance (safety, jerk and velocity)

- What was the tuning process for the loss weights (appendix B2)?

- Is any checking done on biases within the trajectory primitive library (is it balanced, does it average to a straight line from start to goal?)

- How much harder would the authors expect it to be to achieve similar results with RGB camera only?

---

### Note · Authors · 2024-11-23

**Comment:**

We appreciate the feedback from the reviewer.  We fully agree with the reviewer's suggestion that our work should be submitted to conferences or journals related to robotic systems. Therefore, considering that ICLR may not be the most suitable venue for our research, we have decided to withdraw the paper

**Withdrawal Confirmation:**

I have read and agree with the venue's withdrawal policy on behalf of myself and my co-authors.